# Synthesis of Anion Exchange Membranes Containing PVDF/PES and Either PEI or Fumion^®^

**DOI:** 10.3390/membranes12100959

**Published:** 2022-09-30

**Authors:** Luis Javier Salazar-Gastelum, Brenda Yazmin Garcia-Limon, Shui Wai Lin, Julio Cesar Calva-Yañez, Arturo Zizumbo-Lopez, Tatiana Romero-Castañón, Moises Israel Salazar-Gastelum, Sergio Pérez-Sicairos

**Affiliations:** 1Centro de Graduados e Investigación en Química, Instituto Tecnológico de Tijuana, Tecnológico Nacional de México, Blvd. Alberto Limón Padilla, S/N Col. Otay Tecnológico, Tijuana 22510, Mexico; 2Departamento de Ingeniería Eléctrica y Electrónica, Instituto Tecnológico de Tijuana, Tecnológico Nacional de México, Blvd. Alberto Limón Padilla, S/N Col. Otay Tecnológico, Tijuana 22510, Mexico; 3Centro de Graduados e Investigación en Química, Instituto Tecnológico de Tijuana, CONACyT-Tecnológico Nacional de México, Blvd. Alberto Limón Padilla, S/N Col. Otay Tecnológico, Tijuana 22510, Mexico; 4Instituto Nacional de Electricidad y Energías Limpias, Ave. Reforma 113 Col. Palmira, Cuernavaca 62490, Mexico

**Keywords:** alkaline fuel cell, anion exchange membrane, Fumion^®^, blended membrane, phase inversion method, membrane characterization

## Abstract

In this work, the preparation of dense blended membranes, from blends of poly(vinylidene fluoride) (PVDF), poly(ether sulfone) (PES) and polyethyleneimine (PEI) or Fumion^®^, with possible applications in alkaline fuel cell (AEMFC) is reported. The blended PEI/Fumion^®^ membranes were prepared under a controlled air atmosphere by a solvent evaporation method, and were characterized regarding water uptake, swelling ratio, thermogravimetric analysis (TGA), infrared spectroscopy (FT-IR), scanning electron microscopy (SEM), ion exchange capacity (IEC), OH^−^ conductivity and novel hydroxide ion exchange rate (HIER), which is related to the mass transport capacity of the OH^−^ ions through the membrane. The effect of the chemical composition on its morphological and anion exchange properties was evaluated. It was expected that the usage of a commercial ionomer Fumion^®^, in the blended membranes would result in better features in the electrical/ionic conductivity behaviour. However, two of the membranes containing PEI exhibited a higher HIER and OH^−^ conductivity than Fumion^®^ membranes, and were excellent option for potential applications in AEMFC, considering their performance and the cost of Fumion^®^-based membranes.

## 1. Introduction

Due to continuous human population growth, global energy demand increases year by year. Humanity’s energy requirements are closely related to the social, health and economic issues that society demands [1]. During the second half of the 18th century, coal began as the preferred fuel power source for steam engine operation resulting in the world entering the fuel era with the spread of the industrial revolution. Coal produces more heat during combustion compared to biomass-based fuels. The use of coal as fuel was chosen because it was a more affordable option than other choices [2]. Nevertheless, the use of this type of fuel for the usage power generation led to environmental consequences, the most critical of which is climate change, as reported by Peterson [3]. In addition, the depletion of petroleum, natural gas and coal resources has economic impacts. To avoid these issues, governments around the world have adopted new policies and regulations regarding CO_2_ emissions to establish use of renewable energy sources (RES) in national systems, leading to the development of technologies by which different RES can be exploited to satisfy society’s energy demand. The fluctuating nature of different RES represents one of the biggest challenges to overcome in the integration of the RES to the electrical energy network. Unfortunately, RES depends on several conditions, such as location, time, and season, e.g., wind power profiles vary over minutes, hours, and days, while peaking at night when demand is low. During the day, wind power can be a few gigawatts (GW) at some time and only a few megawatts (MW), or even zero, at others. The intermittency of RES results in spatial and temporal gaps between the availability of energy and its consumption by end users. Therefore, reliable energy storage systems must be developed to mitigate these issues [4,5,6,7,8,9]. In Mexico, it is projected for the year 2050 that only 50% of the electric energy will be generated from fossil sources [10]. RES are continually restored by nature, are derived directly from solar energy, or indirectly from wind, or other natural sources such as tidal and geothermal. Unlike fossil fuels, RES does not generate waste, converting this natural energy into forms of usable energy such as electrical and heat energy in a clean manner [11].

It is well known that there are different devices for storage of electrical energy, such as mechanical, chemical and thermal storage [12]. Chemical storage is based on the use of different electrochemical devices to generate electrical energy by two approaches: (i) electrical charge accumulation, such as capacitors and super capacitors, and (ii) by redox reactions such as batteries and fuel cells. The latter electrochemical devices have larger power density in comparison with capacitors and super capacitors, and are ideal devices for portable and stationary technologies [13,14]. A hydrogen fuel cell converts chemical energy directly into electricity, producing only water and heat as by-products, and can supply energy with efficiencies above 80% in contrast with thermal devices that operate under the Carnot cycle [15]. Five types of fuel cells can be distinguished mainly by the type of electrolyte, such as polymeric exchange membrane fuel cells (PEMFC/AEMFC), phosphoric acid fuel cells (PAFC), molten carbonate fuel cells (MCFC) and solid oxide fuel cells (SOFC) [16]. A PEMFC consists of an ionic exchange membrane in contact with two electrodes on both sides. Depending upon the transferred ionic species, the technology is classified as an anion (OH^−^) or proton (H^+^) exchange membrane fuel cell (AEMFC, and PEMFC, respectively). AEMFCs offer the possibility of Pt-free cathode electrodes, with simpler kinetics, a lower fuel crossover and the reduction of CO poisoning. This makes AEMFCs a more affordable technology than PEMFCs. However, PEMFCs have been widely studied, with improvements over AEMFCs such as better water management, higher ionic conductivity and increasing durability in proton exchange membranes. The process of carbonation is one of the main drawbacks of AEMFCs because some CO_2_ from the air is incorporated into the system reacting with OH^−^ producing CO_3_^2−^ and HCO_3_^−^. This decreases anion conductivity in AEM, and these ions are bigger than OH^−^ ion, which reduces AEM conductivity [17,18,19,20,21]. One aspect to consider for AEMFCs is the poor stability of ammonium groups used for hydroxide ion conductivity because this kind of cation is susceptible to react under conditions of the AEMFC by either SN2 or E2 reactions [22,23,24,25].

AEMFC and PEMFC technologies exhibit great versatility, allowing operation with a large number of fuels, such as alcohol, ammonia, and hydrocarbon compounds. Hydrogen is the best option due to carbon-free by-products (with no contribution to the greenhouse effect), which otherwise could cause deactivation of the electrodes attached to the membrane because of the adsorption phenomenon. Hydrogen is continuously fed towards the anode, while the oxygen coming from the air is directly supplied to the cathode. The chemical reactions that take place during operation of the AEMFC are shown below (Equation (1) (hydrogen oxidation reaction, HOR) and Equation (2) (oxygen reduction reaction, ORR), respectively) [26]:(1)$$\mathrm{Anode}\;\mathrm{reaction}:\;2H_{2}+4OH^{-}\to 4H_{2}O+4e^{-}$$
(2)$$\mathrm{Cathode}\;\mathrm{reaction}:\;O_{2}+2H_{2}O+4e^{-}\to 4OH^{-}$$

AEMFC technologies can use a liquid fuel for the oxidation reaction such as ethanol, methanol, glycerol, ethylene glycol and some N-based fuels such as hydrazine, urea and ammonia [27,28,29,30,31,32]. However, AEMFCs using hydrogen as fuel are preferred because they have better OCV potentials and have a minimal environmental impact due to the absence of CO_2_ in the electrochemical reactions [33].

The capacity of OH^−^ transport at the membrane determines the performance of the AEMFC, since the OH^−^ ions must migrate from the cathode to the anode. Chan et al. reported amination in aqueous solutions of different chlorinated polymers via a hydrothermal process, where conventional polymers such as chlorinated polypropylene were functionalized with polyethyleneimine (PEI), obtaining conductivities in the order of 10^−2^ S cm^−1^. Since the membrane exhibited a large porosity, the high conductivity was attributed to functionalization with ammonium groups [34].

Lee et al. [35] prepared membranes with a blend of copolymer stryren-*co*-vinylimidazole and polyvinyl alcohol. These membranes were used for alkaline water electrolysers. The authors reported that a phase separation took place with increased imidazole ratio, yielding hydrophilic channels for ion conductivity.

Wu et al. [36] prepared membranes of quaternized polyether ether ketone (q-PEEK) with a piperidinium cation and polybenzimidazole. The authors reported that all membranes exhibited good miscibility due to a hydrogen bond between the piperidinium cation and N atoms in the main skeleton of polybenzimidazole.

Kerres et al. [37] prepared blended membranes from poly(2,6-dimethyl-1,4-phenyleneoxide). Partially fluorinated polybenzimidazole as the polymer matrix, polyethersulfone was used to create an acid-base crosslinking. Furthermore, the authors used fluorinated polybenzimidazole as an inert polymer to increase mechanical resistance.

Henkensmeier et al. [38] prepared membranes from blends of polybenzimidazole (mPBI) with Fumion^®^ doped with KOH at different mass ratios, finding that all blended membranes exhibited higher values of ionic conductivity than mPBI membranes. One material showed up to 80% higher conductivity. The authors concluded that the weight percent of KOH as a dopant affects all the evaluated properties of membranes. This group studied similar membranes for other devices, such as vanadium redox flow batteries, where the authors observed that blended membranes showed similar results to a commercial membrane (Nafion^®^) [39].

Ramontja et al. [40] studied membranes from quaternized poly(2,6-dimethyl-1,4-phenyleneoxide) and polysulfone (PSf), two rigid polymers due to their aromatic structure. By SEM analysis, the authors found that by increasing the PSf content, the membrane surface showed less imperfections due to better compatibility, yielding a material with great mechanical properties.

In this paper the preparation of different anion exchange membranes containing either Fumion^®^ or PEI as OH^−^ conductive polymers is reported. The content of such ionomers ranged from 3–6% in the blended membranes. Additionally, the feasibility of its potential use as an anion exchange fuel cell solid state electrolyte was established, and as a strategy to reduce the cost of this component in AEMFC by using a lesser quantity of ionomer.

## 2. Experimental Section

### 2.1. Reagents

Pellets of polyvinylidene fluoride (PVDF) (M_W_ = 530,000); polyethyleneimine (PEI, water free; M_W_ = 10,000, *N*-methyl-2-pyrrolidone (NMP, 99.98%) anhydrous and hydrochloric acid (37%) were purchased from Sigma Aldrich. Polyethersulfone (PES, ≥99%; M_W_ = 30,000) was acquired from Imperial Chemical Industry. Fumion^®^ (≈10 wt% solution in NMP) was purchased from Fuel Cell Store. Sodium hydroxide (97%) and FeSO_4_·7H_2_O (>99%) were obtained from Jalmek, phenolphthalein ACS reagent grade and H_2_O_2_ (30%) were purchased from Fagalab.

### 2.2. Membrane Preparation

Polymeric membranes were prepared by a solvent evaporation method using different chemical compositions. PVDF, PES and PEI were dissolved in NMP at room temperature to obtain a solution with 16 wt% of solid content. For the solution containing Fumion^®^, the NMP added was adjusted because this commercial ionomer is already dissolved in the same solvent. The structure of this ionomer is not well known, but has quaternary ammonium groups for hydroxide ion conduction [41]. The casting solution was poured onto a clean glass plate placed in a membrane casting device (for polymeric solution containing PEI it was heated to 60 °C for 2 h in order to minimize the effect of viscosity during casting knife procedure). Afterwards, the glass plate/membrane was placed into a convection oven at 30 °C for 24 h to remove the solvent in a controlled environment. Finally, the membranes were removed from the glass plate by immersing it in DI water at room temperature. The concentration of PEI or Fumion^®^ ranged from 3 to 6 wt%. To study the effect of the polymer on the performance of the membrane, the PES content varied from 0 and 3 wt%, since it should not be in a higher proportion than PEI or Fumion^®^. The PVDF content was fixed at 10 wt% for all membranes. The composition of the prepared membranes is described in Table 1.

#### 2.2.1. Quaternization of PEI/PVDF Membranes

The different PEI functionalized membranes were quaternizated according to the method reported by Eyal and Canari [42], which consists of an acid-base titration, where sufficiently basic amines form H-bonds and cause ion pair formation. Zapata-Gonzalez et al. reported pKa values of polymers with tertiary amines about 7.5 [43]. A slight modification to the Eyal method was used consisting of the substitution of deionized waster by methanol to prepare the 1.0 M HCl. For membranes containing Fumion^®^, this step was not required since the ionomer is in ionic form.

#### 2.2.2. Alkaline Treatment of Membranes

The quaternizated membranes were immersed in a 1.0 M NaOH solution at room temperature for 12 h to replace the hydroxyl (OH^−^) groups at the membrane surface. After this procedure the membranes were washed with DI water until the pH of the remaining solution was neutral. All the membranes were stored in DI water for further characterization.

### 2.3. Membrane Characterization

#### 2.3.1. Fourier Transform Infrared (FT-IR) Spectroscopy

The chemical composition of the membranes surfaces was characterized by FT-IR spectroscopy (Perkin Elmer Spectrum 100) with an attenuated total reflectance device. The membranes were scanned in the range of 600–4000 cm^−1^ with a resolution of 2 cm^−1^ and 16 scans.

#### 2.3.2. Scanning Electron Microscopy (SEM), Energy Dispersive Spectroscopy (EDS) and Atomic Force Microscopy (AFM)

The acquisition of SEM images was carried-out with a microscope (TESCAN, VEGA 3) operated at an accelerating voltage of 15 kV. Previous to analysis, all samples were sputter-coated with gold (99.999%) at 18 mA for 4 min using a SPI-MODULE sputter coater. The presence of polymer blends on the surface membranes were observed semi-quantitatively by energy dispersive *X*-ray spectroscopy (EDS) by mapping measurements on cross-sections using a Bruker XFlash 4010. EDS information was processed using Quantax 200 ESPRIT 1.9 software. AFM analyses were carried out using a microscope (Easyscan 2 Nanosurf) with a scanned area of 100 μm × 100 μm in contact mode.

#### 2.3.3. Thermogravimetric Analysis (TGA)

Thermogravimetric analysis of PEI or Fumion^®^/PVDF/PEI membranes was performed with a TGA Q500 TA Instrument by using a Pt pan. Thermograms were recorded at a temperature range of 30 °C to 800 °C, and a heating rate of 20 °C min^−1^ under a nitrogen atmosphere (60 L min^−1^).

#### 2.3.4. Water Uptake (WU) and Swelling Ratio (SR)

The WU and SR of PEI, Fumion^®^ and PVDF/PES membranes, in OH^−^ form, were calculated by changes of the mass and thickness of the swollen and dried membranes. Each membrane was kept in DI water for 24 h, then excess water was removed with tissue paper and the weight of the wet membrane was recorded and its thickness measured with a Mitutoyo 547–400S. Then, the wet membranes were dried in an oven at 60 °C until constant weight and thickness were measured. WU and SR of membranes were calculated from Equations (3) and (4), respectively:(3)$$WU\left(\%\right)=\frac{m_{wet}-m_{dry}}{m_{dry}}*100\%$$
(4)$$SR\left(\%\right)=\frac{L_{wet}-L_{dry}}{L_{dry}}*100\%$$
where *m_wet_*, *m_dry_*, *L_wet_* and *L_dry_* are the weight and the thickness of the swollen and dried membranes, respectively, and *L_wet_* and *L_dry_* are the thicknesses of fully swollen and dried membranes, respectively.

#### 2.3.5. Ion Exchange Capacity (IEC)

The IEC of membranes was measured using an acid-base titration method as reported in a previous study [44]. The membranes were cut in pieces of 2 cm × 2 cm, then immersed in 10 mL of 0.1 M HCl and kept for 24 h to ensure the consumption of OH^−^ with H^+^. Then, the remaining acidic solution was transferred to a flask and titrated with 0.1 M NaOH using phenolphthalein (1%) as indicator. The IEC of membranes was recorded in meq of H^+^ g^−1^ and was determined by the difference in the number of milliequivalents of HCl solution before (meq of HCl)_0_ and after (meq of HCl)_f_, the membrane being neutralized, according to the Equation (5)
IEC = [(meq of HCl)_0_ − (meq of HCl)_f_]/mass of dry membrane (g)(5)

#### 2.3.6. Hydroxide Ion Conductivity

Anionic conductivity was measured by electrochemistry impedance spectroscopy (EIS) using a four probe H-cell configuration (Figure 1a). Impedance spectroscopy was obtained in a frequency range of 1 MHz-10 Hz with a potential amplitude of 10 mV, using a BioLogic VMP-300 potentiostat. Platinum plates (1 cm × 1 cm) and an Ag/AgCl electrode were used as work and reference electrodes, respectively. NaOH 1.0 M solution was used as the electrolyte. The membranes were cut into squares of 2 cm × 2 cm and placed between the two compartments of the H-type cell. OH^−^ conductivity was calculated by Equation (6):(6)$$\sigma =\frac{L}{R_{mem}A}=\frac{L}{ER}$$
where σ is the hydroxide conductivity in mS cm^−1^, R_mem_ was the difference between the cell with only an electrolyte (blank signal) and a cell with a membrane (R_mem+electrolyte_), *L* is width of membrane in cm, *A* is the contact area of membrane with the electrolyte solution in cm^2^, and ER is the specific electric resistance of the membrane in Ω cm^2^, taking into account the effective area exposed during the measurement [45]. All membranes were measured at room temperature.

#### 2.3.7. Hydroxide Ion Exchange Rate (HIER)

To evaluate the HIER of membranes, a laboratory-made acrylic cell with two compartments was used (Figure 1b). During the test, a sample of the membrane was placed between the two compartments then one compartment was filled up with 250 mL solution of NaOH 2.0 M and the other compartment with 250 mL NaCl 2.0 M in order to keep constant the ionic strength in both compartments and induce mass transport of the OH^−^ through the membrane. A pH electrode was immersed in the NaCl compartment to record the pH every minute for 24 h. All experiments were kept at room temperature and under magnetic stirring to minimize the concentration polarization. All membranes were treated with 2.0 M NaOH for 24 h previous to each experiment. The prepared blended membranes were compared with the commercial membrane, Fumapem^®^. The aim of this measurement was to determine the permeation capacity of the ionic species through the membrane, which is desirable in AEMFCs because the OH^−^ ions generated at the cathodic compartment are consumed in the anodic compartment. Although ion conductivity is a measurement associated with the mobility of the ion, in most experimental designs the transport of the ionic species trough the membrane is not considered.

#### 2.3.8. Oxidative Stability of Membranes

During the operation of AEMFCs, the membranes can suffer from oxidative stress resulting from reactive oxygen species involved in the cathodic reaction. To determine the stability of the polymeric membranes against the oxidative species, a membrane sample of each composition was immersed into Fenton reagent (4% of H_2_O_2_ + 3 ppm of FeSO_4_ at 50 °C) for 120 h, the samples were dried overnight and weighed. The stability of the membrane is related to the weight registered after oxidative stress [46].

## 3. Results and Discussion

### 3.1. Synthesized Membranes

In this work, two series of membranes were developed with different concentrations of ionomer (PEI or Fumion^®^) responsible of OH^−^ transport in the membrane. Hence, four membranes containing PVDF/PES/PEI (labelled M1–M4) and four membranes containing PVDF/PES/Fumion^®^ (labelled F1–F4) were prepared. Table 1 shows the composition and thicknesses of all membranes. The chemical composition refers to that of the polymer solution used to prepare such a membrane.

### 3.2. FT-IR Analysis

Figure 2a shows the FT-IR spectra of the membranes prepared with PVDF/PES/PEI (M1–M4). The peak at 3300 cm^−1^ corresponds to water, which means that at a higher concentration of PEI the surface is more hydrophilic. At 3020 cm^−1^ the signal ascribed to C-H of alkenes appears due to the presence of PES in membranes. The peak at 2980 cm^−1^ is associated to the stretching of aliphatic C-H due to PEI or PVDF. At 1650 cm^−1^ a very clear peak appears for bond the C=C bond. Another characteristic signal related to PVDF is the peak at 1070 cm^−1^ [47], ascribed to the absorption of asymmetrical stretching of CF_2._ The signal at 1144 cm^−1^ [48] is caused by symmetric stretching for S=O. The signal at 1170 cm^−1^ for M1 is split into two small peaks for the case of M2–M4 due to the symmetric stretching of the S=O group. The signal at 1570 cm^−1^, due to the bending of N-H, reveals the presence of PEI in the membrane [49]. Figure 2b shows spectra of the membranes containing Fumion^®^ as ionomer. The specification sheet for FAA-3-SOLUT-10 mentions that it contains aromatic polymers, whose signal appears at 3020 cm^−1^, as indicated by the company. All membranes of this series showed signals for PES and for C-H corresponding to the methylene of PVDF. The peak at 1600 cm^−1^ is due to the C=C bond of PES and polymer aromatics of Fumion^®^. The signal for flexion N-H appears at 1570 cm^−1^ and the asymmetrical stretching of CF_2_ for PVDF appears at 1070 cm^−1^.

### 3.3. Scanning Electron Microscope (SEM), Energy Dispersive Spectroscopy (EDS) and Atomic Force Microscope (AFM)

SEM micrographs are shown in Figure 3 and Figure 4. The analysed surface corresponds to that exposed to the air during preparation. Figure 3 shows SEM images for PEI-based membranes. M1 (Figure 3a) has a more heterogeneous surface that indicates a higher porosity, which could promote the transference of OH^−^ through membrane, and also suggests a higher surface area. M2 and M3 (Figure 3b,c) present a more homogeneous surface, which could result from a better chemical mixing of both polymeric solutions. In the case of M4, a heterogeneous surface similar to M1 can be observed. In the latter, this may due to poor miscibility between PVDF and PEI because PEI is a hydrophilic polymer and PVDF is a hydrophobic polymer due to the C-F bond. M4 has a pore-like structure on the surface, which can be related to the composition solution. This membrane is composed of a mass ratio of PEI/PES equal to 1, probably induced by competition between these two polymers during membrane formation.

SEM images for Fumion^®^-based membranes are shown in Figure 4. These membranes exhibit a more homogenous surface than the PEI-based membranes, and a pore-like pattern is observed in these membranes. With increased concentrations of PES, a greater pore size was detected. Figure 4a shows that the F1 membrane exhibits low porosity and has a homogeneous surface that could be related to good compatibility between PVDF and Fumion^®^. F2 and F3 show repetitive pore-like patterns on the surface, indicating that the casting knife method was suitable for preparing this kind of membrane, depending on the cast solution and type of substrate used. Membrane F4 has a greater pore size compared to other membranes, this is because of competition between Fumion^®^ and PES, showing poor compatibility between these two polymers.

Cross-section micrographs of membranes M1-M4 and F1-F4 are presented in Figure 5. SEM images for M1 and M4 (Figure 5a) reveals that M1 has a structure with a greater porosity than M4. The main parameters favouring this structure are low stability of the components in the polymeric solution, tending to separate, as well as the flux of the solvent evaporating during the preparation process. PES and PEI exhibited less miscibility between them than the Fumion^®^ membranes series. Figure 5b shows SEM images for F1 and F4 where both cross sections are very similar. F4 shows a slightly porous structure due to the incorporation of PES that has the property of creating free spaces due to interaction with PVDF. Although M1 exhibited high apparent porosity, the cross section shown in Figure 5a is around the first 5 µm from the edge, but the total thickness is 60 µm. The cross-section micrographs of all membranes in a wider scale of 20 µm is shown in Appendix A. Appendix A corresponds to PEI-based membranes and Appendix A to Fumion^®^-based membranes. M1 shows the same relatively porous structure at the edge, becoming more tortuous in the middle, and reaching a dense region at higher depths. At this deep region, there are no perceptible transport channels that could allow the crossover of gases during the energy conversion process. This heterogeneous structure (porous at the edge, denser at the center of the membrane) favours the deposition of catalysts, since this increases dispersion and avoids the crossover of gases. Opposite to this behaviour, Fumion^®^-based membranes exhibited a homogeneous and denser structure, where there was a high probability of catalyst agglomeration and a low permeation of gases through the catalytic layer.

EDS analysis of membranes was recorded to verify the chemical composition of the surfaces. Figure 6 shows spectra for the membranes containing PEI, where the constituent elements of each polymer in the membrane was detected. It is important to highlight M4 where sulphur is absent, which is consistent with the chemical composition of casting solution. Another important issue is the presence of nitrogen in all series of membrane.

Similar investigation was done for Fumion^®^-based membranes. The preparation was successful, and the EDS matched with FT-IR analysis. EDS confirmed the presence of elements of each polymer, such as fluorine coming from PVDF, sulphur from PES, and counter ion Br in the case of the commercial ionomer Fumion^®^. In each spectrum there is a signal of this element. F1 in Figure 7a shows the absence of sulphur.

The ratios between N/F and N/C were calculated for all membranes. These elements were selected because N is in the functional group responsible for ion exchange and F and C are present in all polymeric matrices. The results are shown in Table 2 and were taken from the surface.

The N/F ratio in both sets of membranes has the same trend; having a lower value for the membranes with 6 wt% of ionomer, and increasing for membranes with 5 wt%, then decreasing as the ionomer content decreases. This behavior can be related to the morphology at the surface due to the preparation method, although it does not necessarily correspond to the distribution in the internal structure of the membranes.

For evaluation of membrane surfaces, AFM analysis was performed. In Figure 8, M1 and M4 show a rough topography, and SEM images show these two have a higher surface area, indicating that these two membranes are capable of attaching a higher quantity of hydroxide ions. M2 and M3 have a smoother surface (mean roughness of 148 and 169 nm, respectively) in comparison with M1 and M4 (mean roughness of 630 and 454 nm, respectively) indicating that PES has miscibility with PVDF. Nevertheless, an increasing composition of PES promotes competition with PEI, increasing roughness.

Membranes containing Fumion^®^ exhibited a smoother surface (mean roughness ranging between 105 and 345 nm) in comparison with PEI-based membranes (mean roughness ranging between 148 and 630 nm). This is due to better miscibility with those polymers, confirmed SEM images shown before. These membranes have lower surface areas (Figure 9).

### 3.4. Thermogravimetric Analysis (TGA)

The thermal behaviour of membranes was studied by TGA. All the membranes exhibited excellent thermal stability because in the range of 80–150 °C (typical operation range of AEMFC) there was no presence of any degradation step. This is an effect of PVDF, which is a very stable polymer as reported by Lai et al. [50] Figure 10a shows that membranes containing PEI have three important loss stages. The first one at 190 °C is because of the loss of amine groups [51] and the second and third steps are due to degradation of backbone polymers at approximately 500 °C. It is difficult to elucidate which corresponds to each polymer due to the fact that both PES and PVDF degrade at temperatures above 500 °C. In a derivative plot it was observed that a higher quantity of PEI or Fumion^®^ this peak was more evident, which is clear evidence that M1 and F1 have higher values for the amine group.

### 3.5. Water Uptake, Swelling Ratio, Ion Exchange Capacity and Hydroxide Conductivity

Water uptake and swelling ratio are shown in Figure 11. For PEI-based membranes the ability for water uptake is remarkable. That is a useful characteristic because membranes must be moist for amines to make hydrogen bonds. Other membranes, such as M1, M3, and M4, exhibited a low swelling ratio. Fumion^®^-based membranes showed lower water uptake in comparison with PEI membranes and exhibited lower physical deformation.

IEC and hydroxide conductivity values are shown in Table 3. Membranes with a higher PEI/Fumion^®^ content showed higher IEC values (milliequivalents per unit mass). Hydroxide conductivity had a different trend depending on the membrane series. Membranes with a higher content of Fumion^®^ had higher conductivities. Membrane M1 (6 wt% of PEI) showed the highest value of conductivity, but an inversed trend was observed for membranes M2. M3 and M4, increasing the conductivity when the content or PEI decreased in polymeric solution. This trend can be related to the morphological properties of membranes. M4 had higher porosity, which could favour hydroxide conductivity.

### 3.6. Hydroxide Ions Exchange Rate (HIER)

The results of HIER are shown in Figure 12. As shown in Figure 12a, when ratio PEI increases a higher pH value was reached, which is consistent with the quantitative value of hydroxide conductivity due to PEI content. Moreover, M2-M3 showed a similar value of pH, which is in accordance with surface charge density. Regarding analysis of HIER behaviour for short times, M2 reached equilibrium rapidly compared to M4, which can be related to the internal resistance of the membranes to mass and charge transference. For membranes with Fumion^®^, F1 reached a higher pH value compared toF2–F4, but this occurred at a slow rate. For these membranes, F2 had a faster HIER compared with F1, although the latter had higher concentration of ionomer polymer. Figure 12b shows that M4 and F4 have lower pH values.

Figure 13 shows the behavior of HIER for the first 60 min. Different slopes can be observed, which implies different internal resistances of each membrane due to their internal chemical composition, as their morphology and can influence the real performance of the materials because it determines the transference rate of mass and charge across the membranes, among other phenomena.

Table 4 shows the HIER values. Two PEI-synthesized membranes showed a higher value compared to Fumion^®^ membranes, which demonstrates the effect of several parameters of membranes that play very important roles.

When ion diffusion phenomena are involved in the membrane, its internal morphology is influenced so that the concentration difference between both compartments is the driving force during the ion exchange phenomenon. In addition, the intrinsic resistance of the membrane to ion transfer is affected. The observed value was higher in the membranes of M1 and M3 which have good WU, IEC and σ, favoring higher rates of anion exchange. Porosity of membranes, morphology and the internal chemical availability of hydroxide ion-conducting groups are the main factors affecting these materials.

### 3.7. Oxidative Stability

Figure 14a,b shows the stability of PVDF/PES/PEI and PVDF/PES/Fumion^®^ membranes under oxidative stress. It is important to point out that the membranes with the highest ionomer loading (M1 and F1 for PEI and Fumion^®^, respectively) showed low oxidative stability, since there was a 67% and 88% of residual weight. Even though there was no observation of membrane dissolution; the handling of these membranes was not possible, since the manipulation produced cracking of the membrane. This result was expected because (i) lack of PES in the blend produced a weaker structure in the membrane and/or (ii) the hydrophilic nature of the PEI or Fumion^®^ restricted the attack of free radicals in these domains [58]. Another interesting fact is that the weakening effect of the structure was associated with weight loss, which was more evident for the PVDF/PES/PEI membrane series than for the PVDF/PES/Fumion^®^ membrane series, because Fumion^®^ possesses a polyaromatic backbone that is more stable than the aliphatic structure of the PEI [41].

The membranes containing PVDF/PES/PEI had residual weights from 85–96%, where with increasing PES caused a weakening in the structure. This can be explained based on the disaggregation of the polymers in the blend with higher PES content, while a PES content of 1–2% presented practically the same stability (M2 and M3). This disaggregation phenomenon in PVDF/PES/PEI membranes is in accord with the SEM images (Figure 3), showing that poor miscibility between PVDF and PEI producing sponge structure membranes.

On the other hand, either lower disaggregation or no disaggregation was observed in PVDF/PES/Fumion^®^ blends, since the residual weights ranged from 94–97%. The observed trend followed the increasing order of PES content and stability, where F3 and F4 showed similar behaviour.

## 4. Conclusions

The preparation of blended membranes by the solvent evaporation method with different chemical compositions (PVDF/PES/PEI and PVDF/PES/Fumion^®^) is reported in this work. These membranes were characterized by ex situ tests to determine their feasibility as possible alternatives for AEMFC application. The polymers used in the blend, and their nature, determined the performance of blended membranes. Incorporating PEI and Fumion^®^ provided OH- conductivity, and decreasing the quantity of Fumion^®^ reduces membrane cost in comparison to high-cost polymeric membranes. Viscosity changes were a very important factor to consider in the preparation to reduce defects during polymeric-film distribution depending on the polymeric blend. FTIR and EDS analyses confirmed the presence of the different functional groups and characteristic elements such as S, N and F associated with the polymers used to prepare the membranes. Despite having a similar composition, these membranes differed in surface morphologies, where the preparation method and type of ionomer (PEI or Fumion^®^) played an important role.

According to AFM and SEM images, PEI membranes displayed a rough surface attributed to the apparent incompatibility of PEI with PES. These membranes featured a high swelling ratio and water uptake, and the capability of generating a large amount of hydrogen bonds, which is important properties for ionic transport across the membrane, but counterproductive for the membrane’s mechanical and/or thermal stability, establishing thermal decomposition of amino groups at 180 °C.

HIER, determined by a lab-made cell, is a parameter that may lead to establishing interesting ion transport phenomena through the membrane. This is a remarkable feature in AEMFC application and is dismissed in most of related work. It is associated with different parameters involved in anion exchange through the membrane that are not discussed in this paper. Considering performance and cost, membranes M1 and M3 emerged as excellent options for potential applications in AEMFCs, compared to Fumion^®^-based and commercial Fumapem^®^-based membranes.

## Figures and Tables

**Figure 1 membranes-12-00959-f001:**
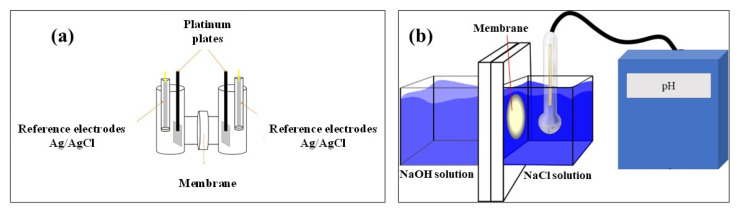
(**a**) H cell configuration for anionic conductivity. (**b**)Two compartment cell for hydroxide ions exchange rate determination.

**Figure 2 membranes-12-00959-f002:**
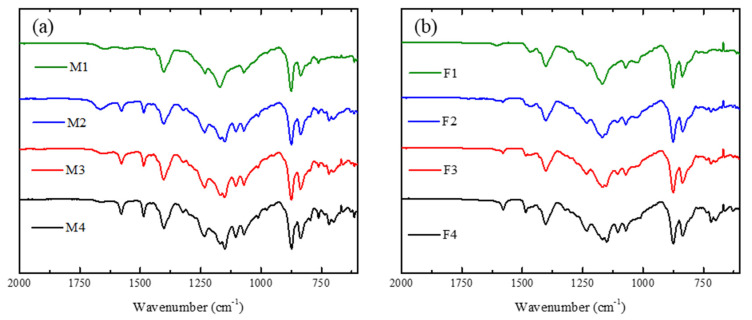
FT-IR spectra of the synthesized membranes containing: (**a**) PEI and (**b**) Fumion^®^.

**Figure 3 membranes-12-00959-f003:**
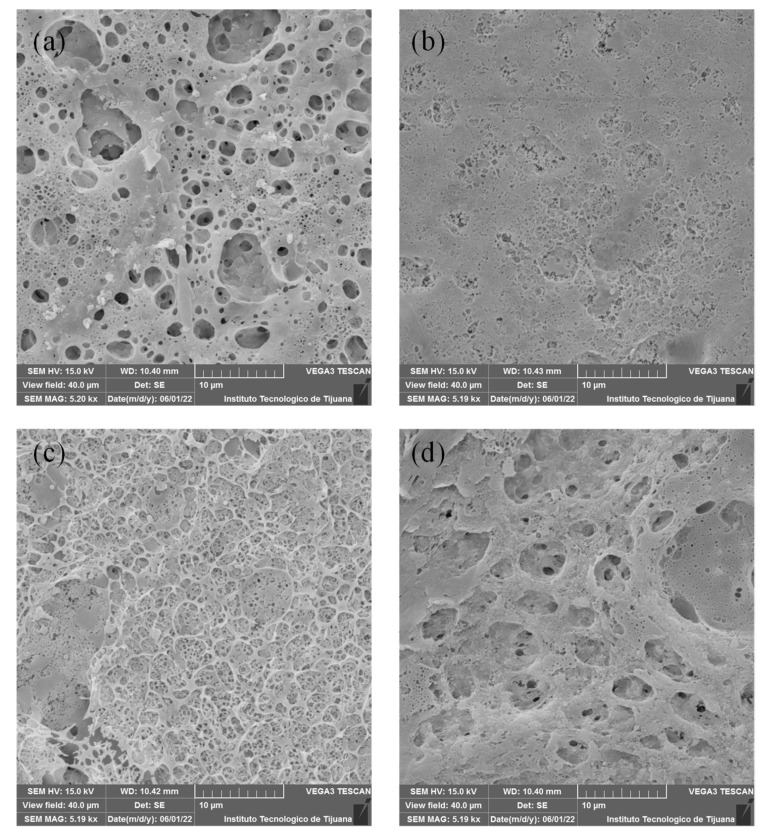
SEM images for membranes containing PEI: (**a**) M1, (**b**) M2, (**c**) M3 y (**d**) M4.

**Figure 4 membranes-12-00959-f004:**
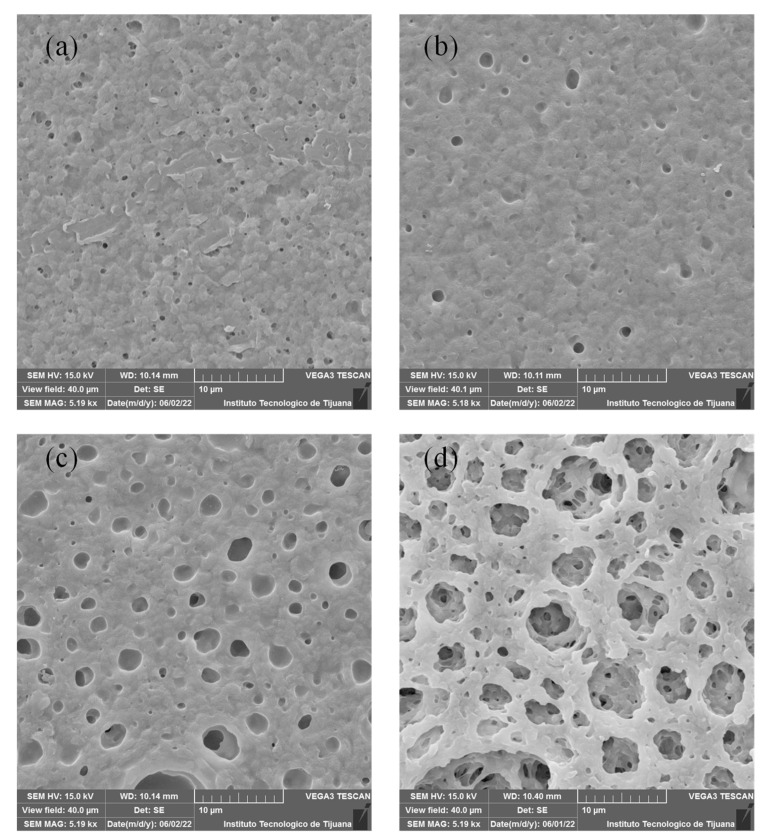
SEM images of membranes containing Fumion^®^: (**a**) F1, (**b**) F2, (**c**) F3 y (**d**) F4.

**Figure 5 membranes-12-00959-f005:**
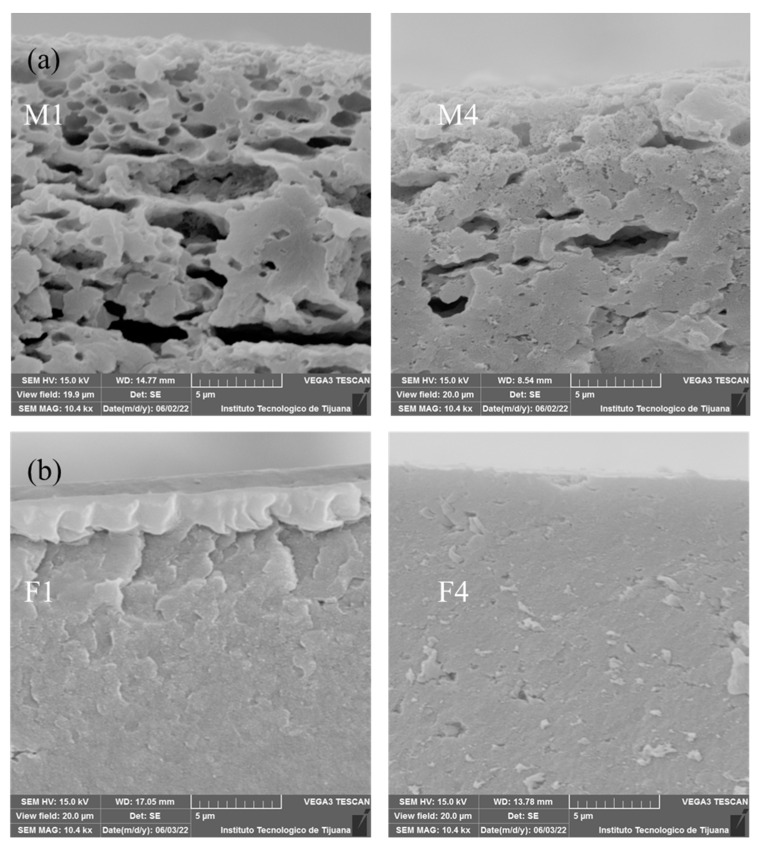
SEM images for membranes: (**a**) M1 and M4 and (**b**) F1 and F4.

**Figure 6 membranes-12-00959-f006:**
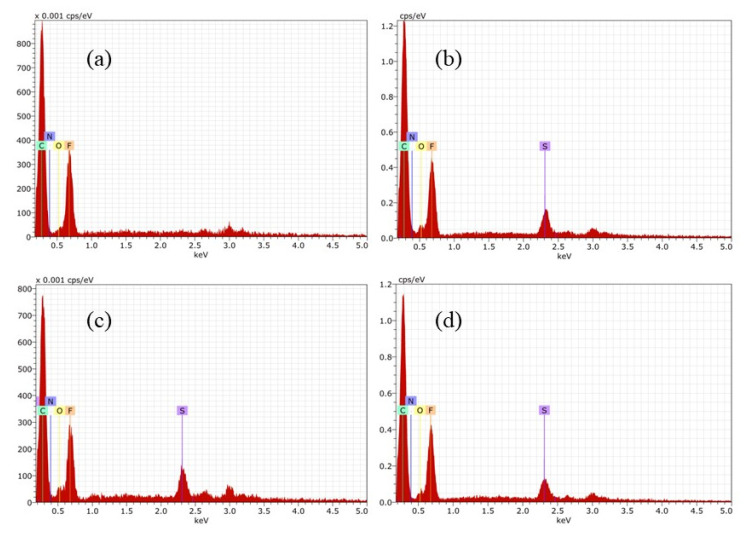
EDS for membranes containing PEI: (**a**) M1, (**b**) M2, (**c**) M3 y (**d**) M4.

**Figure 7 membranes-12-00959-f007:**
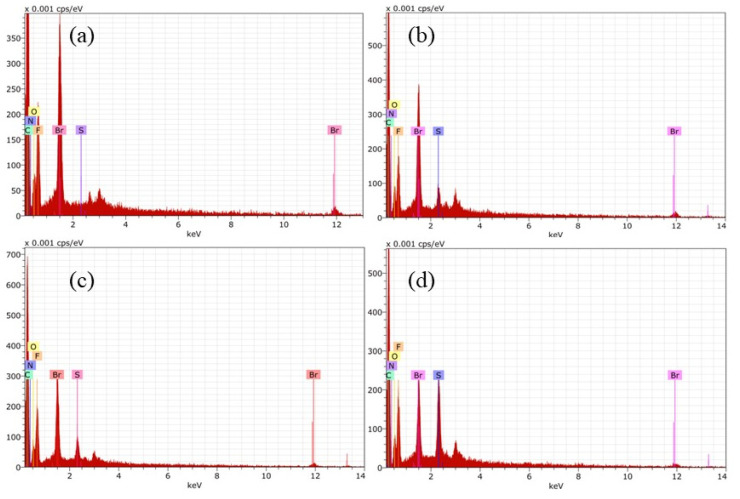
EDS for Fumion^®^-based membranes: (**a**) F1, (**b**) F2, (**c**) F3 y (**d**) F4.

**Figure 8 membranes-12-00959-f008:**
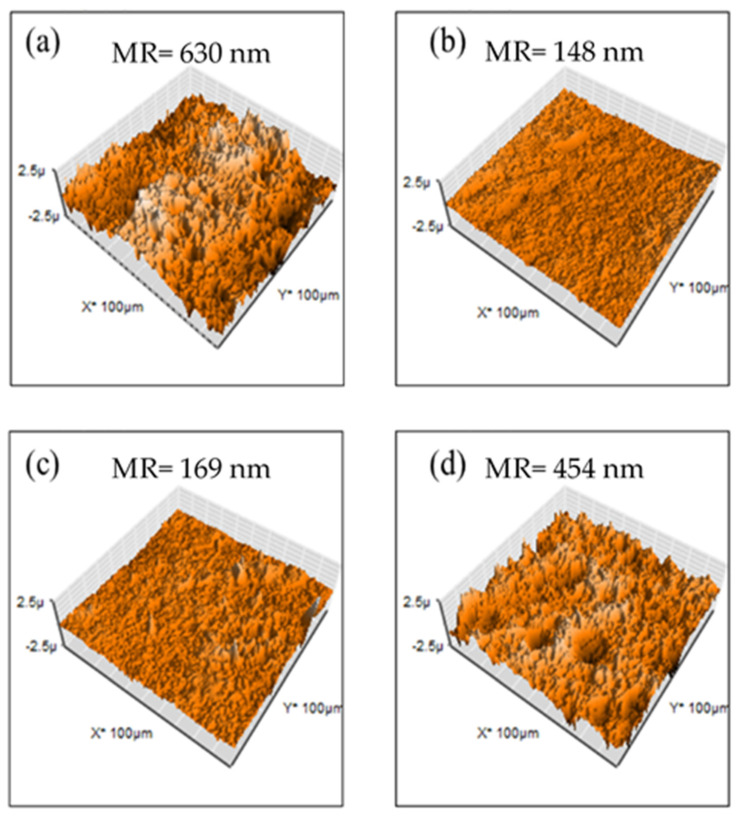
AFM images for PEI-based membranes: (**a**) M1, (**b**) M2, (**c**) M3 y (**d**) M4.

**Figure 9 membranes-12-00959-f009:**
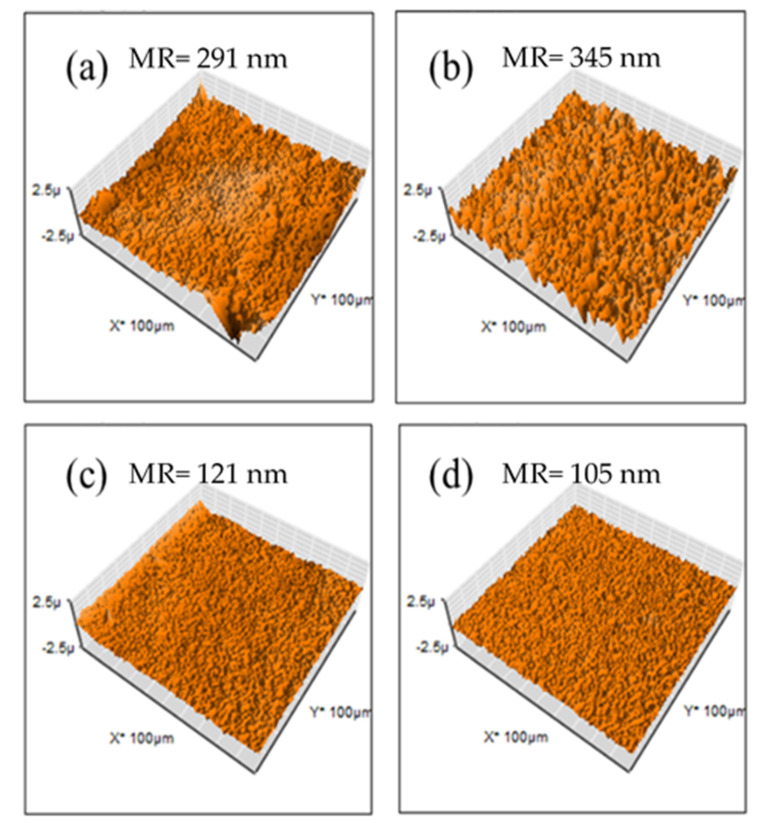
AFM images for Fumion^®^-based membranes: (**a**) F1, (**b**) F2, (**c**) F3 y (**d**) F4.

**Figure 10 membranes-12-00959-f010:**
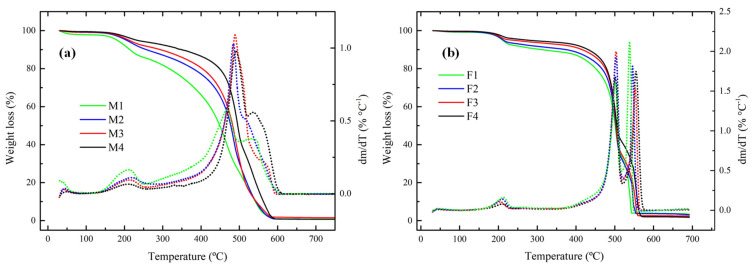
Thermograms for membranes containing: (**a**) PEI and (**b**) Fumion^®^ in N_2_ atmosphere and heating rate of 20 °C min^−1^.

**Figure 11 membranes-12-00959-f011:**
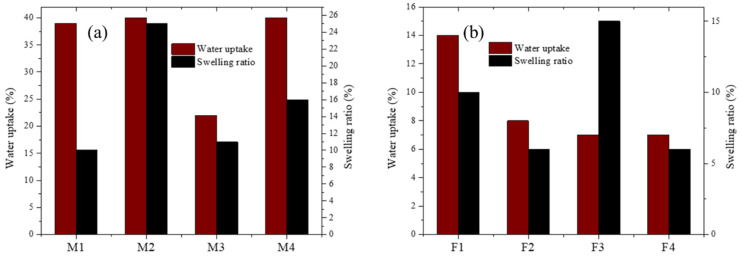
Swelling ratio and water uptake for membranes containing: (**a**) PEI and (**b**) Fumion^®^.

**Figure 12 membranes-12-00959-f012:**
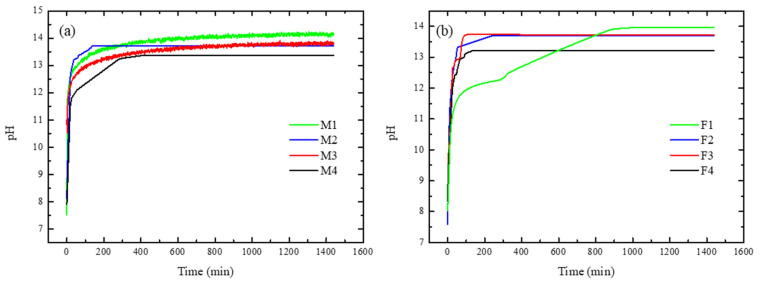
Hydroxide ions exchange rate for membranes containing: (**a**) PEI and (**b**) Fumion^®^.

**Figure 13 membranes-12-00959-f013:**
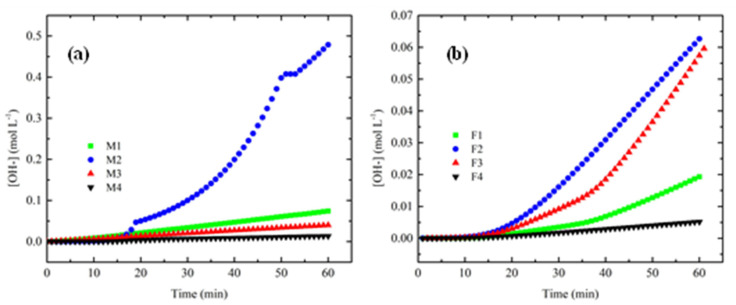
HIER for first 60 min of process for: (**a**) PEI and (**b**) Fumion^®^.

**Figure 14 membranes-12-00959-f014:**
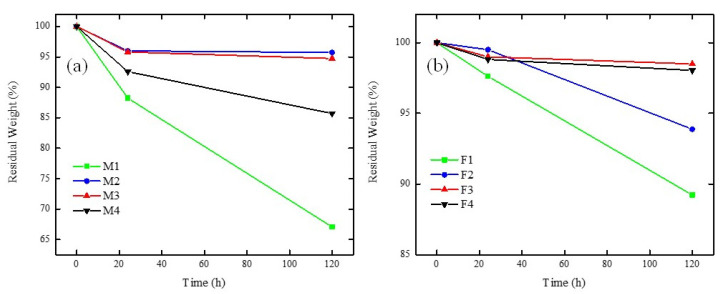
Oxidative stability for membranes containing: (**a**) PEI and (**b**) Fumion^®^ in Fenton reagent at 50 °C.

**Table 1 membranes-12-00959-t001:** Chemical composition and thicknesses of the membranes PVDF/PES and PEI or Fumion^®^.

Name	Chemical Composition	Thickness (μm)	Name	Chemical Composition	Thickness (μm)
M1	PVDF 10%/PES 0%/PEI 6%	60	F1	PVDF 10%/PES 0%/Fumion^®^ 6%	36
M2	PVDF 10%/PES 1%/PEI 5%	30	F2	PVDF 10%/PES 1%/Fumion^®^ 5%	36
M3	PVDF 10%/PES 2%/PEI 4%	41	F3	PVDF 10%/PES 2%/Fumion^®^ 4%	54
M4	PVDF 10%/PES 3%/PEI 3%	35	F4	PVDF 10%/PES 3%/Fumion^®^ 3%	35

**Table 2 membranes-12-00959-t002:** Elemental ratios of N/F and N/C of the synthesized membranes.

Membrane	N (at.%)	F (at.%)	C (at.%)	N/F	N/C	Membrane	N (at.%)	F (at.%)	C (at.%)	N/F	N/C
M1	22.44	17.38	56.58	1.29	0.40	F1	17.61	10.46	62.14	1.68	0.28
M2	22.75	15.00	56.88	1.51	0.40	F2	17.51	9.00	62.63	1.95	0.28
M3	21.72	14.85	56.74	1.46	0.38	F3	16.89	10.61	62.23	1.59	0.27
M4	21.16	15.51	58.22	1.36	0.36	F4	15.78	10.41	62.51	1.51	0.25

**Table 3 membranes-12-00959-t003:** IEC and hydroxide conductivity values for PEI/Fumion^®^ membranes.

Membrane	IEC (meq g^−1^)	σ (mS cm^−1^)	ER (Ω cm^2^)	Ref.
M1	0.316	1.538	2.005 ^a^ [52]	This work
M2	0.076	0.345	7.263 ^a^	
M3	0.044	0.522	7.101 ^a^	
M4	0.034	0.927	3.886 ^a^	
F1	0.144	1.419	2.533 ^a^	
F2	0.080	0.615	5.855^5 a^	
F3	0.050	0.237	21.062 ^a^	
F4	0.049	0.004	875.895 ^a^	
Polybenzimidazol	0.960	0.91	8.300	[53,54]
Neosepta	1.250	b	>50.000	[53,55]
Sustainion 37–50	b	0.080	0.045	[53,56]
AMI-7001	6.000	1.300	2.000	[53,57]

^a^ ER procedure was performed according to Jaime-Ferrer et al. [52]. b Not reported.

**Table 4 membranes-12-00959-t004:** Values of hydroxide ion exchange rate.

Membrane	HIER ($\frac{OH^{-}}{cm^{2}*µm*min}$)
M1	7.45 × 10^16^
M2	1.56 × 10^16^
M3	8.56 × 10^16^
M4	1.68 × 10^15^
F1	1.26 × 10^15^
F2	6.27 × 10^15^
F3	4.59 × 10^15^
F4	2.58 × 10^15^
FUMAPEM^®^	7.53 × 10^15^

## Data Availability

Data is contained within the article.

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
