# Peer review of "Synthesis of Anion Exchange Membranes Containing PVDF/PES and Either PEI or Fumion®"

_membranes, 2022, doi:10.3390/membranes12100959_

Round 1

Reviewer 1 Report

In this manuscript, the preparation of dense composite membranes, from blends of PVDF, PES, PEI, and Fumion®, with possible applications in AEMFC is reported. Several analytical techniques were used for membrane characterization. The composite PEI/Fumion® membranes exhibited a higher hydroxide ions exchange rate than Fumion® membranes, emerging as an excellent option for potential applications in AEMFC.

The contents are interesting and make great sense. I may give a minor revision due to further improvements that are needed by addressing my comments made below.

(1) For the Keywords, ‘Fumion®’, ‘composite membrane’, and ‘membrane characterization’ should be added to attract a broader readership.

(2) Line 44, ‘The fluctuating nature of the different RES represents one of the most challenges to overcome in the integration of the RES to the electrical energy network.It should be explained better, especially the reason and the effect of the fluctuating. For example, most renewable energy sources are intermittent, opening spatial and temporal gaps between the availability of the energy and its consumption by the end-users. In order to address these issues, it is necessary to develop suitable energy storage systems for the power grid [Electrochimica Acta 309 (2019): 311-325].

(3) Line 67, ‘However, AEMFC offers the possibility of Pt-free cathode electrodes, a simpler kinetic behavior, a lower fuel crossover and the reduction of CO poisoning, those facts make AEMFC a more affordable technology than PEMFC.

In terms of catalyst cost, fuel crossover, and avoiding CO poisoning, it is true that AEMFC is better. But how about the operation's current densities? Generally, protons are smaller than anions in terms of ionic size, hence the proton conductivity in PEMs is generally higher than the anionic conductivity in AEMS. This will definitely limit the operational current densities of AEMFC. I suggest giving a comprehensive comparison between PEMFC and AEMFC, not only showing the advantages of AEMFC [Journal of Power Sources 375 (2018): 170-184].

(4) Line 72, ‘Hydrogen reveals as the best option due to the free carbon intermediates, which could cause the deactivation of the electrodes attached to the membrane because of adsorption phenomenon.Hydrogen is also better than other fuels, in terms of CO2 emissions that can be avoided, that is why it is also a very green technique to be used. This point should also be addressed.

(5) Line 109, ‘In this paper, the preparation of different anion exchange membranes using Fumion® and PEI as ionomers is reported.There is no strict one-to-one correspondence between this and the article title ‘membranes containing PVDF/PES/PEI and Fumion®’. It is also clear that in samples F1-F4, there is no PEI at all. So the final part of the Introduction does not well summarize the whole contents of this manuscript.

(6) For Part 2.1, the molecular weight of PEI and PES is still missing. And what is the approximate chemical structure or at least the functional groups of Fumion® should be introduced briefly?

(7) For Part 2.2, refers only to ‘PVDF, PES and PEI were dissolved in NMP at room temperature’, but how is Fumion added and mixed with them? In addition, ‘The concentration of PEI/Fumion® was varied from 3 to 6 wt% in order to study the effect of the polymer on the performance of the membranes. The PVDF content was fixed to 10 wt% for all the study.So how about the amount of control of PES? This information is still missing.

(8) Line 202 and 218, ‘Error! Reference source not found’ have appeared twice. I suggest the authors double-check the whole manuscript to avoid these errors due to the reference insertion software.

(9) The TGA result is not well explained. Only two temperature is mentioned, but the real thermal events taking place should be more complex, since the composite contains different polymer backbones. And what leads to the thermal events different among various composite membranes, this should also be compared and briefly discussed. See the examples of TGA analysis [Solid State Ionics 319 (2018): 110-116], the first derivative of the wt% in the specific range may be very helpful for the result analysis.

(10) For Table 2, I suggest adding some commercial AEM data. Only in this way, the advantage or disadvantage of the prepared membranes can be understood by comparison with the already existing commercial AEMs.

(11) In the conclusion, it refers to ‘Membranes M1 and M3 emerge as an excellent option for potential applications in AEMFC, considering their performance and the cost regarding based Fumion® membranes’. Hence, it is a pity that no in-situ AEMFC single-cell test result is shown in this manuscript, which should be direct proof of the above-mentioned physical-chemical characterizations. Since so many detailed characterizations have been presented, why the single-cell test/polarization curve test of the AEMFC is missing? I also do not consider the device test result can be rewritten as another independent paper.

(12) How about the chemical stability of the obtained membranes? Some Fenton reagents may be used for this kind of test.

Reviewer 2 Report

1. The authors may check other references related properties of anion exchange membrane. The experimental data from the authors's study did not show some good results, it's much worse than current published papers. 

2. It's suggested the authors to read  Kean Long Lim *, Chun Yik Wong, Wai Yin Wong, Kee Shyuan Loh, Sarala Selambakkannu, Nor Azillah Fatimah Othman, Hsiharng Yang *, “Radiation grafted anion exchange membrane for fuel cell and electrolyzer applications: a mini review,” Membranes, 202111(6), 397, 21 pages; https://doi.org/10.3390/membranes11060397.  Once the authors have good enough properties, then the paper will be good for publishing.

Reviewer 3 Report

This work reported the preparation of dense composite membranes, from blends of PVDF, PES, PEI, and Fumion® with possible applications in an alkaline fuel cell. Although the novelty of the membranes in this work is not high, this work established the feasibility of its potential use as an anion exchange fuel cell solid-state electrolyte. It may contribute to related research. However, it looks like an unfinished draft. For example, many “Error! Reference source not found” in the text. Normally, a high attitude should be in the writing when someone submits their work. I suggest it could not be accepted by the Membranes. Other details,

1. Too many paragraphs in the Introduction

2. Could you provide more details on the Hydroxide ions conductivity measurement? For example, how to connect the EIS and four-probe H-cell configuration. A picture or drawing may useful.

3. Line 202, “In order to evaluate the HIER of membranes a laboratory-made acrylic cell with two compartments (Error! Reference source not found.) was used.” What do you mean by the “Error! Reference source not found”? Also in many other places, such as line 218 “Error! Reference source not found. shows the composition and thicknesses of all the membranes reported in this work.” Line 249 “Error! Reference source not found. shows SEM images for the membranes containing PEI.” Line 252 “In the other hand M2 and M3 (Error! Reference source not found. (b) y (c)) presents”, etc. Please check the whole manuscript carefully!

4. The Conclusions are too long and not clear!

Round 2

Reviewer 3 Report

1. Some references should be added in part 3.2. FTIR analysis.

2. Please provide the ratio of the elements in Figure 6 and Figure 7.

3. Please provide the roughness of the membranes in Figure 8 and in Figure 9. 

4. Figure 11. The swelling ratio and water uptake are different meanings. You cannot show them by using the same Y-axial.

5. The authors shouldn't make a discussion in "Part 4 Conclusion". I suggest providing a short conclusion to make a clear conclusion of this work! 

Round 3

Reviewer 3 Report

Please carefully check the English writing.